# Characterization and Antibacterial Activity of Silver Nanoparticles Synthesized from *Oxya chinensis sinuosa* (Grasshopper) Extract

**DOI:** 10.3390/microorganisms12102089

**Published:** 2024-10-18

**Authors:** Se-Min Kim, Tai-Yong Kim, Yun-Sang Choi, Gyeongsik Ok, Min-Cheol Lim

**Affiliations:** 1Research Group of Food Safety and Distribution, Korea Food Research Institute, Wanju-gun 55365, Republic of Korea; k.semin@kfri.re.kr (S.-M.K.); k.taiyong@kfri.re.kr (T.-Y.K.); gsok@kfri.re.kr (G.O.); 2Department of Food Science and Biotechnology, Chung-Ang University, Anseong-si 17546, Republic of Korea; 3Research Group of Food Processing, Korea Food Research Institute, Wanju-gun 55365, Republic of Korea; kcys0517@kfri.re.kr; 4Department of Food Biotechnology, Korea University of Science and Technology, Daejeon-si 34113, Republic of Korea

**Keywords:** silver nanoparticles, green synthesis, edible insects, *Oxya chinensis sinuosa*, AgNP characterization, antibacterial activity

## Abstract

In this study, silver nanoparticles (AgNPs) were synthesized using a green method from an extract of the edible insect *Oxya chinensis sinuosa* (O_extract). The formation of AgNPs (O_AgNPs) was confirmed via UV–vis spectroscopy, and their stability was assessed using Turbiscan analysis. The size and morphology of the synthesized particles were characterized using transmission electron microscopy and field-emission scanning electron microscopy. Dynamic light scattering and zeta potential analyses further confirmed the size distribution and dispersion stability of the particles. The average particle size was 111.8 ± 1.5 nm, indicating relatively high stability. The synthesized O_AgNPs were further characterized using X-ray photoelectron spectroscopy (XPS), high-resolution X-ray diffraction (HR-XRD), and Fourier transform infrared (FTIR) spectroscopy. XPS analysis confirmed the chemical composition of the O_AgNP surface, whereas HR-XRD confirmed its crystallinity. FTIR analysis suggested that the O_extract plays a crucial role in the synthesis process. The antibacterial activity of the O_AgNPs was demonstrated using a disk diffusion assay, which revealed effective activity against common foodborne pathogens, including *Salmonella* Typhimurium, *Escherichia coli*, *Staphylococcus aureus*, and *Bacillus cereus*. O_AgNPs exhibited clear antibacterial activity, with inhibition zones of 15.08 ± 0.45 mm for *S*. Typhimurium, 15.03 ± 0.15 mm for *E. coli*, 15.24 ± 0.66 mm for *S. aureus*, and 13.30 ± 0.16 mm for *B. cereus*. These findings suggest that the O_AgNPs synthesized from the O_extract have potential for use as antibacterial agents against foodborne bacteria.

## 1. Introduction

Foodborne illnesses remain a significant global health concern, with millions of cases reported annually. Among various causes of these illnesses, the most prevalent ones are bacterial infections resulting from contaminated foods [1]. Some foodborne bacteria can cause illness even at very low concentrations and, in severe cases, can be fatal. Contamination can occur at multiple stages, ranging from the handling of raw ingredients to packaging and distribution [2]. Therefore, preventing contamination and ensuring early detection are crucial. It is equally important to control microorganisms when the contamination levels are below the detectable threshold. This has led to growing interest in integrating antibacterial agents into food packaging materials.

Nanoparticles, defined as particles smaller than 100 nm, are emerging as promising materials for food packaging because of their unique physicochemical properties, such as large surface area, charge, and dispersibility [3]. Among the various nanoparticles, silver nanoparticles (AgNPs) have garnered the most attention. AgNPs are used in biomedicine, agriculture, food packaging, cosmetics, catalysis, and sensors owing to their antibacterial properties [4]. Their multifunctional properties, including antibacterial, antifungal, antiviral, and antioxidant properties, make them particularly attractive for incorporation into food packaging materials [5,6]. Notably, AgNPs offer strong antibacterial activity with low toxicity to animal cells, further enhancing their potential for safe use in food applications [7,8,9,10,11].

Nanoparticle synthesis methods, including those for AgNPs, can be broadly classified into two approaches: top–down and bottom–up [12]. The top–down approach involves the reduction of bulk materials to nanoparticles through processes such as chemical etching, mechanical milling, laser cutting, lithography, thermal decomposition, and sputtering [13,14]. Although effective, this approach can lead to surface defects, high energy consumption, and high cost. In contrast, the bottom–up approach involves synthesizing nanoparticles from atoms using either chemical or biological methods. Chemical synthesis is widely used for metal nanoparticle production, providing precise control over particle properties. However, the chemicals involved and the by-products generated can pose risks to both human health and the environment [15]. Thus, there is a growing demand for eco-friendly reducing agents to replace traditional organic and inorganic ones. Although AgNPs are generally safe for use in human and animal cells, they pose environmental and health risks. This has spurred interest in greener biological synthesis methods that utilize bacteria, fungi, plant extracts, and polysaccharides [16,17]. Biological methods are advantageous because they avoid toxic chemicals, simplify the manufacturing process, and shorten reaction times [18]. However, challenges, such as maintaining sterile conditions and culturing cells, render microorganism-based synthesis less practical, leading to a preference for plant extracts [19]. Although plant-based syntheses have been widely studied, there is still a need to explore and develop other cost-effective and nontoxic biomaterials in addition to plants.

Amid growing environmental concerns, there is an increasing global imperative to develop eco-friendly materials. Demand for alternative proteins to reduce carbon emissions and ensure food security has increased in the food industry. Among various alternative protein sources, edible insects offer attractive solutions because of their nutritional and environmental benefits. Edible insects grow rapidly, require minimal space, and are cost-effective [20,21]. They are rich in proteins, fats, minerals, and vitamins. In Korea, seven species of edible insects are currently approved, namely, *Gryllus bimaculatus*, *Oxya chinensis sinuosa*, *Locusta migratoria*, *Tenebrio molitor* larvae, *Zophobas atratus* larvae, *Protaetia brevitarsis seulensis* larvae, and *Trypoxylus dichotomus* larvae. Among these, *Oxya chinensis sinuosa* (grasshopper) has been consumed as a food for centuries and has a long history of medicinal use in Korea [22]. Dried *Oxya chinensis sinuosa* contains 3% fat, 74% protein, 18% carbohydrates, 4% ash, and various minerals, such as Ca, P, Mg, Zn, Fe, Cu, Mn, B, and Mo [23]. Previous studies have demonstrated that biomolecules, such as amino acids, vitamins, proteins, enzymes, and polysaccharides, from biological extracts can act as reducing and stabilizing agents in synthesizing AgNPs [12]. Recently, the use of insects to utilize proteins and amino acids for synthesizing AgNPs has also been reported [20,21,22,23,24]. These findings suggested that the abundant proteins, vitamins, and polyphenols in *Oxya chinensis sinuosa* may serve as effective catalysts for synthesizing AgNPs. As an edible insect, *Oxya chinensis sinuosa* offers a safer alternative to nonedible insects, making it a promising biomaterial for the green synthesis of AgNPs.

In this study, edible *Oxya chinensis sinuosa* was selected as a new, eco-friendly material for the synthesis of AgNPs with low toxicity to humans. The aim was to synthesize AgNPs using the entire *Oxya chinensis sinuosa*, rather than just by-products or specific parts. The synthesized AgNPs were comprehensively characterized using UV–vis spectroscopy, transmission electron microscopy (TEM), field-emission scanning electron microscopy (FE-SEM), dynamic light scattering (DLS), zeta potential analysis, X-ray photoelectron spectroscopy (XPS), high-resolution X-ray diffraction (HR-XRD), and Fourier transform infrared (FTIR) spectroscopy. Additionally, their antibacterial activity against common foodborne pathogens was evaluated using disk diffusion assays. The findings of these analyses highlight the potential applications of the synthesized AgNPs as antibacterial agents in food packaging materials.

## 2. Materials and Methods

### 2.1. Materials

Freeze-dried powder (O_powder) samples of *Oxya chinensis sinuosa* were obtained from a local market (Farmbang, Republic of Korea) and stored at 4 °C until the experiment. Furthermore, a 0.1 M NaOH solution to prepare an extract of *Oxya chinensis sinuosa* (O_extract), 1 N HCl for pH adjustment, and silver nitrate were purchased from Sigma-Aldrich (St. Louis, MO, USA). Tryptic soy broth (TSB) and Tryptic soy agar (TSA) were purchased from BD Biosciences (Sparks, MD, USA) to confirm the antibacterial activity of the synthesized AgNPs. A 10 kDa MWCO centrifugal filter device (Amicon Ultra-15, Millipore, Burlington, MA, USA) was used to wash the synthesized AgNPs.

### 2.2. O_extract Production and AgNP Synthesis

The extraction conditions were determined through preliminary experiments. Briefly, O_powder amounts of 0.1, 0.3, and 0.5 g were tested, and the absorbance values of the AgNPs synthesized from each extract were compared to determine the optimal ratio between the insect powder and 0.1 M NaOH. Based on the method described by Jakinala et al., O_powder (0.5 g) was mixed with 20 mL of 0.1 M NaOH solution in a vial [25]. The mixture was heated in an oven at 90 °C for 1 h with intermittent mixing every 20 min. Then, AgNPs (O_AgNPs) were synthesized by mixing 49 mL of the pH-adjusted extract with 1 mL of 0.1 M AgNO₃ solution. After allowing the reaction to proceed for 48 h at room temperature, 20 mL of the supernatant was collected and filtered through a filter tube via centrifugation at 7200× *g* for 10 min. The filtered O_AgNPs were washed three times with distilled water (DW) and dispersed in DW. The synthesized O_AgNPs were stored at 4 °C until further characterization and antibacterial activity assays were conducted.

### 2.3. UV–Vis Spectroscopy Analysis

UV–vis absorption spectra were obtained to determine the formation of AgNPs and monitor changes over the reaction time. The absorbance of the synthesized solutions was recorded over the wavelength range of 300–700 nm using a SpectraMax i3X microplate reader (Molecular Devices, Sunnyvale, CA, USA).

### 2.4. Dispersion Stability Analysis

The stability of the synthesized AgNPs was assessed using a Turbiscan Tower (Formulaction, L’Union, France). The samples were placed in clear vials, and the system measured the transmittance and backscattering of near-infrared light (λ = 880 nm) at intervals of 40 μm from the bottom to the top of the sample. These measurements were obtained every minute over 3 h at a constant temperature of 25 °C to evaluate the stability of the dispersion.

### 2.5. TEM Analysis

The morphology and size of the AgNPs were examined using a biological transmission electron microscope (Bio-TEM, Cerritos, CA, USA) equipped with an H-7650 electron microscope (Hitachi, Tokyo, Japan) at an operating voltage of 120 kV. Samples were prepared by placing a drop of the nanoparticle solution onto a carbon 200 mesh copper grid (Ted Pella, Redding, CA, USA) and drying overnight in a desiccator before analysis using Bio-TEM.

### 2.6. FE-SEM Analysis

The morphology of the produced AgNPs was further analyzed using FE-SEM (SUPRA 40VP, Carl Zeiss, Jena, Germany) operated at 15 kV. For this analysis, each sample was drop-cast onto a silicon wafer and dried overnight in a desiccator.

### 2.7. Particle Size and Surface Charge Analysis

The particle size distribution and surface charge of the synthesized AgNPs were analyzed using DLS (Zetasizer Nano ZS90, Malvern Instruments, Malvern, UK). The particle size and zeta potential were measured at 25 °C with a wavelength of 660 nm.

### 2.8. XPS Analysis

The surface characteristics and binding energies of the synthesized AgNPs were examined using XPS. Colloidal AgNP samples were deposited onto a silicon wafer and analyzed with an AXIS Supra+ (Kratos Analytical, Stretford, UK) to determine the composition and bonding states.

### 2.9. HR-XRD Analysis

The crystallinity of the AgNPs was analyzed using HR-XRD (Empyrean, PANalytical, Houston, TX, USA) within a 2θ range of 10° to 80°, utilizing Cu Kα radiation (λ = 0.15418 nm) at 40 kV and 40 mA. For this analysis, the samples were drop-cast onto glass slides and dried overnight in a desiccator.

### 2.10. FTIR Analysis

FTIR analysis was used to identify the functional groups present in the synthesized AgNPs. The differences between the O_extract and O_AgNP samples were analyzed using an FTIR spectrometer (6300FV+IRT-5000, JASCO, Easton, MD, USA). For FTIR analysis, the washed O_AgNPs sample was freeze-dried to obtain a powder. The powdered O_AgNPs were then pressed into KBr pellets, and the spectra were recorded in the range of 400–4000 cm^−1^.

### 2.11. Antibacterial Activity

This study aimed to evaluate the antibacterial efficacy of the O_AgNPs synthesized from the O_extract against foodborne pathogens. The optimal conditions were determined through preliminary experiments based on the method described by Kim et al. [26]. Representative foodborne bacteria, including Gram-negative strains (*Salmonella* Typhimurium ATCC 14028 and *Escherichia coli* ATCC 10536) and Gram-positive strains (*Staphylococcus aureus* ATCC 25923 and *Bacillus cereus* KCTC 1094), were selected for testing. All of these strains are non-antibiotic-resistant, allowing for a clear comparison of the antibacterial activity with standard antibiotics. A disk diffusion assay was used to assess the antibacterial activity of these strains. All the bacteria were cultured overnight in 5 mL of TSB at 37 °C. Each culture was diluted to 10^7^ CFU/mL, swabbed onto a TSA plate using a sterile cotton swab, and then dried. After drying, seven sterilized cellulose disks were placed on each plate. To accurately confirm the antibacterial activity of the O_AgNPs, two disks of washed AgNPs were loaded onto one plate after synthesis. To confirm whether the O_extract itself had antibacterial activity, two disks were loaded onto one plate in the same manner as the O_AgNPs. For the comparison, DW and 1 mM AgNO_3_ were used as negative controls. Kanamycin (30 μg), an antibiotic that acts on all strains, was used as a positive control. After placing 30 μL of each sample on sterilized cellulose disks and incubating them at 37 °C for 24 h, the inhibition zone was measured.

## 3. Results and Discussion

### 3.1. Confirmation of Synthesis of AgNPs

The process of synthesizing O_AgNPs from the O_extract is shown in Figure 1A. The synthesis of O_AgNPs was first confirmed visually by the color change in the reaction solution. As shown in Figure 1B, when 1 mM AgNO_3_ was mixed with the O_extract, the color of the solution changed from translucent to brown after the reaction. According to Krishnaraj et al., this color change is due to the excitation of surface plasmon resonance in the metal particles, indicating the formation of AgNPs [27]. UV–vis absorption spectra were measured to further verify the synthesis of AgNPs (Figure 1C). After the synthesis reaction, the absorbance of the solution was measured in the wavelength range of 300–700 nm. Previous studies have shown that the green synthesis of AgNPs exhibits the highest absorbance peak in the range of 400–450 nm [28,29,30]. The UV–vis absorption spectrum of the O_AgNPs showed a maximum peak at 408 nm, indicating the reduction in Ag^+^ to AgNPs (Ag^0^) [31]. To monitor the progress of the synthesis of AgNPs and the changes in absorbance over time, absorbance was measured at different time points (Figure 1C). With prolonged synthesis time, the overall absorbance increased, and the peaks became more pronounced, confirming that the AgNPs synthesized from the O_extract were stable over time. The high dispersion stability of the synthesized O_AgNPs was further confirmed via Turbiscan analysis, which provided additional evidence of stability along with absorbance measurements (Appendix A).

In the green synthesis of AgNPs, plant extracts are often used as reducing agents because of their steroid, sapogenin, carbohydrate, and flavonoid contents [32]. Previous studies have shown that *Oxya chinensis sinuosa* contains a significant number of polyphenols, which are presumed to act as reducing agents in the synthesis of AgNPs [33]. The results of the Turbiscan analysis and UV–vis measurements indicated that the AgNPs synthesized from the O_extract exhibited significant stability over an extended period, suggesting that the O_extract is a viable material for synthesizing AgNPs.

### 3.2. Size and Shape Analysis of Synthesized AgNPs

TEM and FE-SEM analyses were performed on the washed samples to determine the morphological characteristics of the O_AgNPs synthesized using the O_extract. The TEM images (Figure 2A) show that the synthesized O_AgNPs were relatively well dispersed and exhibited small spherical shapes. The results of FE-SEM (Figure 2B) were highly consistent with those of TEM. The spherical shape is one of the defining characteristics of AgNPs that are primarily synthesized through the green synthesis method [24,27,29]. The smaller and more evenly distributed the AgNPs produced, the larger the surface area, which facilitates higher antibacterial activity against pathogenic microorganisms [34].

The hydrodynamic size, surface zeta potential, and polydispersity index (PDI) values of the synthesized O_AgNPs were measured via DLS. The particle characterization results for the O_AgNPs are summarized in Table 1. The generated O_AgNPs showed a relatively even particle size distribution, and the average particle size was approximately 111.8 ± 1.5 nm (Figure 2C). The difference in the average particle sizes between the TEM/FE-SEM and DLS measurements may be attributed to the fact that TEM and FE-SEM analyze only a portion of the sample, whereas DLS considers the entire particle population. The negative zeta potential of the O_AgNPs suggests electrostatic repulsion between particles, which contributes to the dispersion and stability of the nanoparticles in colloidal solutions [35,36]. A previous study has shown that the larger the absolute value of the zeta potential, the more stable the nanoparticles are, which helps prevent aggregation [37]. The zeta potential of the O_AgNPs was −24.1 ± 0.9 mV, which exceeds the threshold of 20 mV, indicating relatively high stability. The PDI indicates the particle size distribution of the O_AgNPs, where 0 indicates a monodisperse distribution and 1 indicates a polydisperse distribution [38]. The closer the value is to 0, the less varied and more consistent is the particle size distribution. The PDI of the synthesized O_AgNPs was 0.226, indicating a monodisperse distribution. Thus, the O_AgNPs exhibited relatively high stability with a small size and uniform distribution, which is consistent with the results described in Section 3.1.

### 3.3. Characterization of the Synthesized AgNPs

#### 3.3.1. XPS Analysis

The chemical composition of the surface of the O_AgNPs was analyzed using XPS. Characteristic peaks were observed at each location in the XPS survey spectrum (Figure 3A). These results indicate the presence of C, O, N, and Ag on the surface of the O_AgNPs. Figure 3B shows the XPS Ag 3d scan spectra of the O_AgNPs. Two specific peaks at 367.7 eV and 373.4 eV were observed in the Ag 3d region. These peaks are associated with spin–orbit splitting, corresponding to the Ag 3d_5/2_ and Ag 3d_3/2_ core levels, respectively [39,40]. The peak difference between the Ag 3d_5/2_ and Ag 3d_3/2_ core levels was approximately 6 eV, indicating the presence of the Ag^0^ state in the generated O_AgNPs and the successful formation of AgNPs from the O_extract [41].

#### 3.3.2. HR-XRD Analysis

The crystal characteristics of the O_AgNPs were analyzed using HR-XRD. The XRD pattern of the AgNPs synthesized using the O_extract is shown in Figure 3C. Peaks were observed at 38.4°, 46.4°, and 77.1°, corresponding to the (111), (220), and (311) crystal planes of the AgNPs, respectively, with a face-centered cubic structure. Notably, the weak peak at 38.4° (111) correlated with the plane of the Ag crystal, indicating that the O_AgNPs were produced in a crystalline form [42]. The additional peaks at 27.9°, 32.3°, 55.0°, 57.7°, and 67.6° correspond to the (111), (200), (311), (222), and (400) planes of AgCl, respectively [43]. This suggested the presence of chloride ions in the O_extract, which might have led to the formation of AgCl nanoparticles [44]. Therefore, XRD analysis suggests that the O_AgNP solution synthesized from the O_extract may contain a mixture of crystalline AgNPs and potentially some AgCl nanoparticles.

#### 3.3.3. FTIR Analysis

The functional groups of the synthesized O_AgNPs were identified using FTIR analysis. Figure 3D shows the FTIR spectra of the O_extract and O_AgNPs in the spectral range of 500–4000 cm^−1^. The synthesized O_AgNPs showed evident peaks at 3286 cm^−1^, 2933 cm^−1^, 2856 cm^−1^, 1652 cm^−1^, 1538 cm^−1^, 1384 cm^−1^, 1247 cm^−1^, 1081 cm^−1^, and 832 cm^−1^, which were confirmed to be similar to the spectrum of the O_extract (Table 2). The peaks at 3286 cm^−1^ (O-H stretching vibration) and 2933 cm^−1^ (C-H stretching vibration) correspond to the aliphatic hydrocarbon groups of polyphenols, polysaccharides, and proteins bound to the AgO surface [45]. The peak at 1652 cm^−1^ is attributed to the C=O stretching vibration. The C-H peak and C=O peak intensities increased compared with those of the O_extract. These results suggest that both hydroxyl and carbonyl groups are involved in the reduction and stabilization of O_AgNPs [36]. The peak at 832 cm^−1^, which was observed only in the O_AgNPs, is related to the C-H stretching vibration of the alkene group, along with the peak at 2933 cm^−1^ [46].

Moreover, as indicated by the colored markers in Figure 3D, the peaks at 1538 cm^−1^, 1384 cm^−1^, and 1247 cm^−1^ were interpreted as peaks related to amino acids. The weak peaks around 1538 cm^−1^ and 1247 cm^−1^ represent the amide II and III bonds between the amino acid residues present in the protein, respectively [43]. The peak at 1384 cm^−1^ is attributed to the C-N asymmetric stretching vibration associated with the aromatic ring, and the peak at 1081 cm^−1^ corresponds to the C-O stretching vibration [47].

By comparing the spectra of the O_AgNPs and O_extract, shifts and changes in the intensities of some peaks were observed. This suggests that the O_extract is involved in the synthesis of AgNPs. When the overall spectrum was confirmed, the O_extract was observed to contain proteins, polyphenols, and carboxylic acids and is believed to act as a stabilizing and dispersing agent during the synthesis of AgNPs.

### 3.4. Antibacterial Activity of Synthesized AgNPs

One of the most significant characteristics of AgNPs is their antibacterial activity against various bacteria [48]. The development of a green synthesis method for AgNPs is a promising approach for the control of foodborne pathogens in the food packaging industry. Accordingly, representative foodborne bacteria, including Gram-negative *S*. Typhimurium and *E. coli* and Gram-positive *S. aureus* and *B. cereus*, were selected for analysis.

The antibacterial activity of the synthesized O_AgNPs was confirmed using the disk diffusion assay (Figure 4). For a comparison, kanamycin (30 μg), O_extract, and 1 mM AgNO_3_ solution were also tested on the same plate. The inhibition zones were measured as follows: *S*. Typhimurium, 15.08 ± 0.45 mm; *E. coli*, 15.03 ± 0.15 mm; *S. aureus*, 15.24 ± 0.66 mm; and *B. cereus*, 13.30 ± 0.16 mm (Table 3). Compared with kanamycin (30 μg), which was used as a positive control, the O_AgNPs produced a clear inhibition zone, indicating significant antibacterial activity against both Gram-negative and Gram-positive bacteria.

The O_extract was tested separately to confirm whether the antibacterial activity was due to the O_extract. The absence of an inhibition zone in all four strains indicated that the O_extract did not exhibit antibacterial activity against the selected bacteria. Additionally, a 1 mM AgNO_3_ solution was evaluated. Although the 1 mM AgNO_3_ solution exhibited minimal antibacterial activity, the inhibition zones for the O_AgNPs were markedly larger across all four strains, indicating that the synthesized AgNPs had more potent antibacterial effects.

All the bacteria were cultured under identical conditions. *B. cereus* is a Gram-positive bacterium with a thick cell wall primarily composed of peptidoglycan and is known to form spores and biofilms [48]. Due to its strong defensive mechanisms against extreme conditions, it is presumed to exhibit higher resistance to antibacterial agents compared to not only Gram-negative bacteria but also other Gram-positive bacteria such as *S. aureus*. This observation is consistent with findings from previous studies [49]. Notably, the *B. cereus* strain used in this experiment consistently exhibited larger and thicker colony growth compared to other strains when cultured at the same temperature, which may have hindered the diffusion of O_AgNPs, reducing their antibacterial activity. Among the bacteria tested, S. aureus showed the most pronounced sensitivity to the O_AgNPs. The antibacterial activity of AgNPs may involve several mechanisms, such as direct interaction with the cell wall, generation of reactive oxygen species (ROS), and release of metal ions [50,51]. These mechanisms disrupt the bacterial cell wall and membrane, leading to structural deformation, leakage of intracellular contents, and DNA damage. In addition to physical effects, silver ions released from AgNPs likely contribute to the antibacterial activity. Previous studies have suggested that the antibacterial activity is due to electrostatic interactions between silver ions and bacteria [49,52]. However, the exact mechanism of the antibacterial activity caused by AgNPs remains unclear, necessitating further investigation. In conclusion, the synthesized O_AgNPs demonstrated significant antibacterial activity against the four foodborne pathogens tested.

Many previous studies have attempted to incorporate antimicrobial AgNPs into bio-degradable and non-biodegradable polymers for food packaging and have shown effectiveness in improving food safety and shelf life [53,54,55,56]. However, human toxicity is a major hurdle for the successful introduction of AgNPs as an antipathogenic agent into food packaging. Although the probability is low, AgNPs transferred from packaging to food may accumulate and exhibit toxicity in various organs in the body [57,58,59]. Therefore, additional processing (coating and heat treatment) of packaging has been reported to reduce the migration of silver nanoparticles into food [60,61]; additional research is still needed to reduce human toxicity due to AgNP exposure.

## 4. Conclusions

This study demonstrated the green synthesis of AgNPs from extracts of the edible insect *Oxya chinensis sinuosa*. The particle characteristics of the synthesized O_AgNPs were analyzed using various characterization techniques, and their antibacterial activity against foodborne pathogens was confirmed. The synthesis was initially indicated by a color change in the O_extract and 1 mM AgNO_3_ reaction solution and further confirmed by the appearance of a maximum peak at 408 nm in the UV–vis absorption spectra. The increasing peak intensity over time and the results from the Turbiscan analysis suggested the high dispersion stability of the synthesized O_AgNPs. TEM and FE-SEM analyses revealed that the O_AgNPs were spherical and well dispersed, which is consistent with the DLS and zeta potential results. The average particle size was approximately 111.8 ± 1.5 nm, with a relatively uniform size distribution. Further characterization using XPS, HR-XRD, and FTIR spectroscopy confirmed the presence of crystalline AgNPs and the role of the O_extract in the synthesis process. The antibacterial activity of the O_AgNPs was evaluated using a disk diffusion assay, which demonstrated their effectiveness against both Gram-positive and Gram-negative foodborne bacteria. In conclusion, this study confirmed the successful synthesis of AgNPs from the O_extract, resulting in relatively small, spherical, highly dispersible, and stable particles. Additionally, the synthesized O_AgNPs exhibited significant antibacterial activity against four types of foodborne pathogens. These findings suggest that the synthesized O_AgNPs have the potential to be used as antimicrobial agents in food packaging, although further optimization studies are required.

## Figures and Tables

**Figure 1 microorganisms-12-02089-f001:**
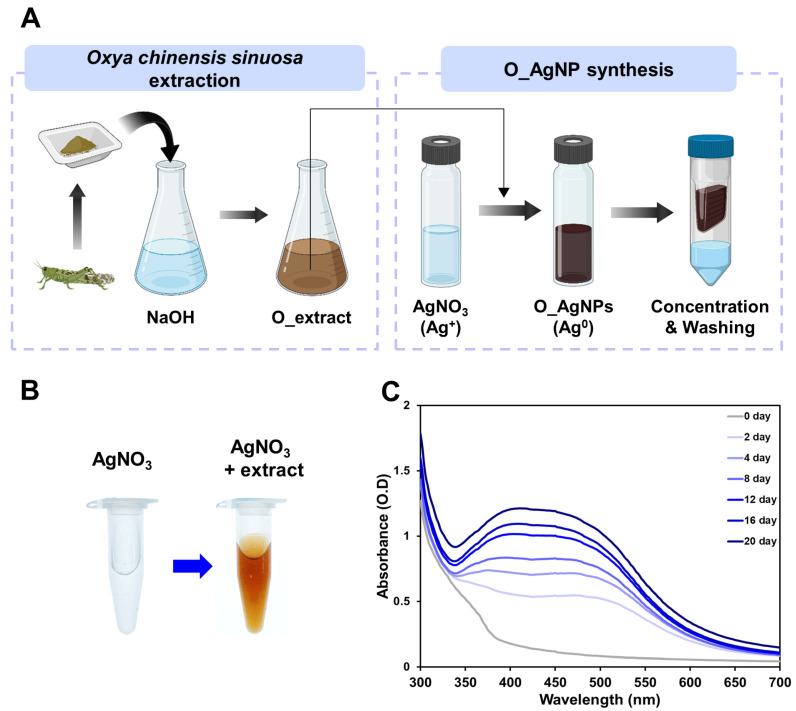
Confirmation of AgNP synthesis. (**A**) Schematic illustration of the synthesis process of O_AgNPs. The figure presented was created with biorender.com. (**B**) Color change in the solution before and after the AgNP synthesis. (**C**) Time-dependent UV–vis absorption spectra for the AgNP synthesis using the O_extract and 1 mM AgNO_3_ solution.

**Figure 2 microorganisms-12-02089-f002:**
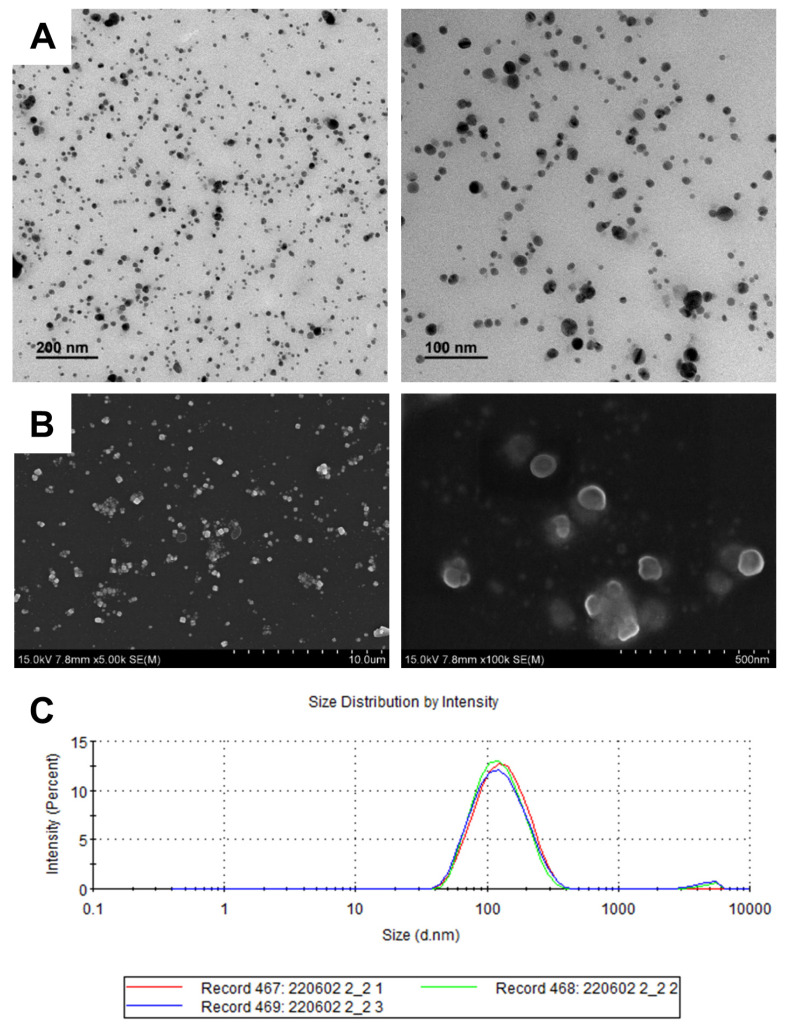
Morphological analysis of the synthesized AgNPs. (**A**) TEM images (different scale bars 200 nm and 100 nm); (**B**) size distribution of the O_AgNPs; FE-SEM analysis of the O_AgNPs. (Different scale bars 10 μm and 500 nm); (**C**) average particle size of O_AgNPs.

**Figure 3 microorganisms-12-02089-f003:**
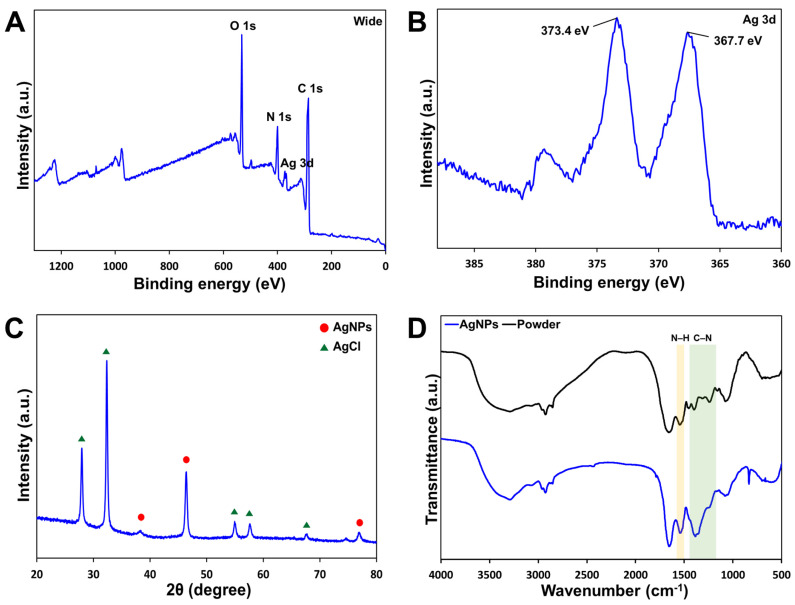
Characterization of the synthesized AgNPs. (**A**) XPS survey scan spectra of the O_AgNPs; (**B**) XPS spectra of the Ag 3d core level of the O_AgNPs; (**C**) HR-XRD pattern; (**D**) FTIR spectra of the O_powder and O_AgNPs.

**Figure 4 microorganisms-12-02089-f004:**
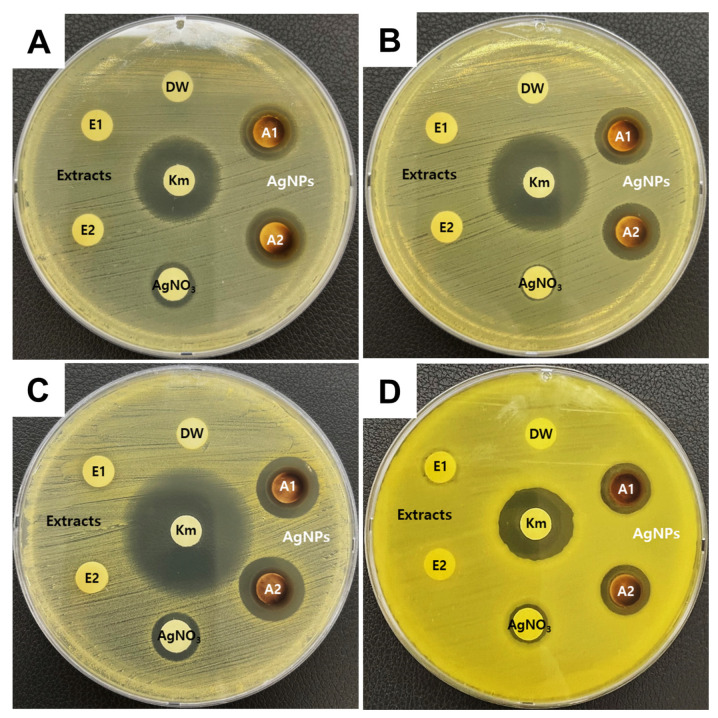
Assessment of the antibacterial activity of the O_extract and O_AgNPs using the disk diffusion assay. (**A**) *S*. Typhimurium, (**B**) *E. coli*, (**C**) *S. aureus*, and (**D**) *B. cereus*. Both the extract and AgNPs were tested in duplicate on a single plate. DW, distilled water; E, O_extract; A, O_AgNPs; Km, kanamycin.

**Table 1 microorganisms-12-02089-t001:** Size and surface charge characteristics of the O_AgNPs using DLS and zeta-potential analysis.

Source of AgNP Synthesis	Mean Size (nm)	Zeta Potential (mV)	Polydispersity Index (PDI)
O_extract	111.8 ± 1.5	−24.1 ± 0.9	0.226

**Table 2 microorganisms-12-02089-t002:** The peaks are found in the FTIR spectra of the O_extract and O_AgNPs.

Wavenumber (cm^−1^)		Assignment
O_AgNPs	O_extract	
3286	3284	O–H stretching vibration
2933	2929	C–H stretching vibration
1652	1648	C=O stretching vibration
1538	1552	N–H bending (amide II band)
1384	1392	C–N stretching vibration
1247	1234	C–N stretching vibration (amide Ⅲ band)
1081	1070	C–O stretching vibrations
832	–	C–H stretching vibration

**Table 3 microorganisms-12-02089-t003:** Antibacterial activity of the O_extract and O_AgNPs.

Sample	*S*. Typhimurium	*E. coli*	*S. aureus*	*B. cereus*
	Zone of Inhibition (mm)			
O_extract	NA	NA	NA	NA
O_AgNPs	15.08 ± 0.45	15.03 ± 0.15	15.24 ± 0.66	13.30 ± 0.16
AgNO_3_	12.19 ± 0.6	9.78 ± 0.43	10.81 ± 1.26	9.83 ± 0.14
Antibiotics	22.91 ± 0.13	26.45 ± 0.49	30.93 ± 0.27	18.5 ± 0.71

NA, No activity; Values represent the mean of three replicates ± SD.

## Data Availability

Data will be made available on request.

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
