# Peer review of "Characterization and Antibacterial Activity of Silver Nanoparticles Synthesized from *Oxya chinensis sinuosa* (Grasshopper) Extract"

_microorganisms, 2024, doi:10.3390/microorganisms12102089_

Round 1
Reviewer 1 Report
Comments and Suggestions for Authors
Characterization and Antibacterial Activity of Silver Nanoparticles Synthesized from *Oxya chinensis sinuosa* Extract
General Overview:
The manuscript presents a comprehensive study on the green synthesis of silver nanoparticles (AgNPs) using an extract from the edible insect *Oxya chinensis sinuosa*. The research focuses on the characterization of the nanoparticles and the evaluation of their antibacterial activity. The study is methodologically sound, and the results indicate a promising potential for these nanoparticles in food packaging as antimicrobial agents.
- Abstract:
The abstract is clear and provides a good summary of the research. It describes the methodology, key findings, and potential applications. A minor suggestion would be to include quantitative results from the antibacterial tests in the abstract for more immediate clarity.
-Introduction:
A more critical discussion of the limitations of chemical synthesis methods (e.g., environmental hazards) versus biological methods could strengthen the rationale for using insect extracts.
Highlight the novelty of this research.
Materials and Methods:
Some preliminary data referenced (e.g., preliminary experiments for extraction conditions) are not shown. Including this data or explaining its absence might enhance transparency.
It would be beneficial to mention how the bacterial strains were selected and whether these represent the most common or most resistant foodborne pathogens.
Results and Discussion:
The discussion on the stability and size distribution of nanoparticles could be improved by comparing these results to other studies on green synthesis methods, particularly those using insect extracts. A direct comparison with plant-based or other green methods would provide more context on the novelty of the method.
The results for *B. cereus* showed relatively low inhibition compared to the other strains. A more detailed discussion of possible reasons (e.g., thicker biofilm or cell wall) and suggestions for improvement in nanoparticle design would add depth.
While the study effectively shows antibacterial activity, it lacks a discussion on the mechanism of action. Expanding on how AgNPs interact with bacterial cells (e.g., via membrane disruption or generation of reactive oxygen species) would strengthen the mechanistic understanding.
Check these paper:
https://doi.org/10.1186/s12934-024-02484-0
https://doi.org/10.3390/biology10020137
Author Response
Reviewers’ comments:
Reviewer #1
The manuscript presents a comprehensive study on the green synthesis of silver nanoparticles (AgNPs) using an extract from the edible insect *Oxya chinensis sinuosa*. The research focuses on the characterization of the nanoparticles and the evaluation of their antibacterial activity. The study is methodologically sound, and the results indicate a promising potential for these nanoparticles in food packaging as antimicrobial agents.
Comment 1-1:
The abstract is clear and provides a good summary of the research. It describes the methodology, key findings, and potential applications. A minor suggestion would be to include quantitative results from the antibacterial tests in the abstract for more immediate clarity.
Response 1-1:
Thank you for your valuable feedback on the abstract. In response to your suggestion, we have added specific numerical data from the antimicrobial tests to enhance clarity and provide more precise results.
Added in line 28-30, page 1 in the revised manuscript.
O_AgNPs exhibited clear antibacterial activity, with inhibition zones of 15.08 ± 0.45 mm for S. Typhimurium, 15.03 ± 0.15 mm for E. coli, 15.24 ± 0.66 mm for S. aureus, and 13.30 ± 0.16 mm for B. cereus.
Comment 1-2:
- A more critical discussion of the limitations of chemical synthesis methods (e.g., environmental hazards) versus biological methods could strengthen the rationale for using insect extracts.
- Highlight the novelty of this research.
Response 1-2:
Thank you for your thoughtful insights on the introduction. Based on your suggestions, we have revised part of the introduction as follows.
Revised sentence in line 62-66, page 2 in the revised manuscript.
Chemical synthesis is widely used for metal nanoparticle production, providing precise control over particle properties. However, the chemicals involved and the by-products generated can pose risks to both human health and the environment [15]. Thus, there is a growing demand for eco-friendly reducing agents to replace traditional organic and inorganic ones.
Revised sentence in line 96-99, page 3 in the revised manuscript.
In this study, edible Oxya chinensis sinuosa were selected as a new, eco-friendly material for the synthesis of AgNPs with low toxicity to humans. The aim was to synthesize AgNPs using the entire Oxya chinensis sinuosa, rather than just by-products or specific parts.
Comment 1-3:
Some preliminary data referenced (e.g., preliminary experiments for extraction conditions) are not shown. Including this data or explaining its absence might enhance transparency.
Response 1-3:
Thank you for your insightful suggestion. We have briefly described the preliminary experiment process and incorporated the changes into the manuscript.
Revised sentence in line 118-121, page 3 in the revised manuscript.
The extraction conditions were determined through preliminary experiments. Briefly, O_powder amounts of 0.1, 0.3, and 0.5 g were tested, and the absorbance values of the AgNPs synthesized from each extract were compared to determine the optimal ratio between the insect powder and 0.1 M NaOH (data not shown).
Comment 1-4:
It would be beneficial to mention how the bacterial strains were selected and whether these represent the most common or most resistant foodborne pathogens.
Response 1-4:
Thank you for your valuable feedback. In response to your suggestion, we have clarified the criteria for selecting the bacterial strains in the manuscript.
Revised sentence in line 177-181, page 4 in the revised manuscript.
Representative foodborne bacteria, including gram-negative strains (Salmonella Typhimurium ATCC 14028 and Escherichia coli ATCC 10536) and gram-positive strains (Staphylococcus aureus ATCC 25923 and Bacillus cereus KCTC 1094), were selected for testing. All of these strains are non-antibiotic-resistant, allowing for a clear comparison of the antibacterial activity with standard antibiotics.
Comment 1-5:
- The results for * cereus* showed relatively low inhibition compared to the other strains. A more detailed discussion of possible reasons (e.g., thicker biofilm or cell wall) and suggestions for improvement in nanoparticle design would add depth.
- While the study effectively shows antibacterial activity, it lacks a discussion on the mechanism of action. Expanding on how AgNPs interact with bacterial cells (e.g., via membrane disruption or generation of reactive oxygen species) would strengthen the mechanistic understanding.
Response 1-5:
Thank you for your detailed feedback. In full agreement with your suggestions, we have added the relevant information on B. cereus as well as the antibacterial mechanism of AgNPs, along with the corresponding references, as shown below.
Additionally, during the revision process, a numerical error was identified and has been corrected in the manuscript and Table 3.
Revised sentence in line 324-326, page 10 in the revised manuscript.
The inhibition zones were measured as follows: S. Typhimurium, 15.08 ± 0.45 mm; E. coli, 15.03 ± 0.15 mm; S. aureus, 15.24 ± 0.66 mm; and B. cereus, 13.30 ± 0.16 mm (Table 3).
Table 3 in the original manuscript
|
Sample |
S. Typhimurium |
E. coli |
S. aureus |
B. cereus |
|
Zone of Inhibition (mm) |
||||
|
O_extract |
NA |
NA |
NA |
NA |
|
O_AgNPs |
15.08 ± 0.45 |
15.03 ± 0.15 |
14.88 ± 1.03 |
13.30 ± 0.16 |
|
AgNO3 |
12.19 ± 0.6 |
9.78 ± 0.43 |
10.81 ± 1.26 |
9.83 ± 0.14 |
|
Antibiotics |
22.91 ± 0.13 |
26.45 ± 0.49 |
30.93 ± 0.27 |
18.5 ± 0.71 |
Table 3 in the revised manuscript
|
Sample |
S. Typhimurium |
E. coli |
S. aureus |
B. cereus |
|
Zone of Inhibition (mm) |
||||
|
O_extract |
NA |
NA |
NA |
NA |
|
O_AgNPs |
15.08 ± 0.45 |
15.03 ± 0.15 |
15.24 ± 0.66 |
13.30 ± 0.16 |
|
AgNO3 |
12.19 ± 0.6 |
9.78 ± 0.43 |
10.81 ± 1.26 |
9.83 ± 0.14 |
|
Antibiotics |
22.91 ± 0.13 |
26.45 ± 0.49 |
30.93 ± 0.27 |
18.5 ± 0.71 |
Revised sentence in line 344-363, page 11 in the revised manuscript.
All the bacteria were cultured under identical conditions. B. cereus is a Gram-positive bacterium with a thick cell wall primarily composed of peptidoglycan and is known to form spores and biofilms [48]. Due to its strong defensive mechanisms against extreme conditions, it is presumed to exhibit higher resistance to antibacterial agents compared to not only Gram-negative bacteria but also other Gram-positive bacteria such as S. aureus. This observation is consistent with findings from previous studies [49]. Notably, the B. cereus strain used in this experiment consistently exhibited larger and thicker colony growth compared to other strains when cultured at the same temperature, which may have hindered the diffusion of O_AgNPs, reducing their antibacterial activity. Among the bacteria tested, S. aureus showed the most pronounced sensitivity to the O_AgNPs. The antibacterial activity of AgNPs may involve several mechanisms, such as direct interaction with the cell wall, generation of reactive oxygen species (ROS), and release of metal ions [50,51]. These mechanisms disrupt the bacterial cell wall and membrane, leading to structural deformation, leakage of intracellular contents, and DNA damage. In addition to physical effects, silver ions released from AgNPs likely contribute to the antibacterial activity. Previous studies have suggested that the antibacterial activity is due to electrostatic interactions between silver ions and bacteria [49,52]. However, the exact mechanism of the antibacterial activity caused by AgNPs remains unclear, necessitating further investigation. In conclusion, the synthesized O_AgNPs demonstrated significant antibacterial activity against the four foodborne pathogens tested.
Added references in line 516-528, page 15 in the revised manuscript.
Granum, P.E.; Lindbäck, T. Bacillus cereus. In Food Microbiology: Fundamentals and Frontiers 2012, 491–502. DOI:10.1128/9781555818463.ch19.
Metryka, O.; Wasilkowski, D.; Dulski, M.; Adamczyk-Habrajska, M.; Augustyniak, M.; Mrozik, A. Metallic nanoparticle actions on the outer layer structure and properties of Bacillus cereus and Staphylococcus epidermidis. Chemosphere 2024, 354, 141691. DOI:10.1016/j.chemosphere.2024.141691.
Fan, X.; Yahia, L.; Sacher, E. Antimicrobial Properties of the Ag, Cu Nanoparticle System. Biology 2021, 10, 137. DOI:10.3390/biology10020137.
El-Sapagh, S.H.; El-Zawawy, N.A.; Elshobary, M.E.; et al. Harnessing the power of Neobacillus niacini AUMC-B524 for silver oxide nanoparticle synthesis: optimization, characterization, and bioactivity exploration. Microb. Cell Fact. 2024, 23, 220. DOI: 10.1186/s12934-024-02484-0.
Reviewer 2 Report
Comments and Suggestions for Authors
The manuscript entitled “Characterization and Antibacterial Activity of Silver Nanoparticles Synthesized from Oxya chinensis sinuosa (Grasshopper) Extract » is devoted to silver nanoparticles synthesize using a green method its characterization and antibacterial activity assessment against common foodborne pathogens, including Salmonella Typhimurium, Escherichia coli, Staphylococcus aureus, and Bacillus cereus. The experiments were conducted at a high methodological level, which does not allow us to doubt the results obtained. To my mind this manuscript is very topical and contains a lot of important and new data. It corresponding to the aims and scopes of the “Microorganisms” journal. I have no complaints about the scientific content of the material. I am ready to recommend it for publication after correcting several comments.
1. It is worth indicating the conditions of TEM and SEM shooting
2. It is worth indicating how the samples for FTIR were prepared
3. The methods are generally described very sparingly
4. I would advise moving the description of the synthesis method from the results to the methods section
5. The quality of Figure 2 is low, it looks like raw data, it should be moved to the supplementary
6. Redo the legend to Figure 3C, the captions should be clearer
Author Response
Reviewer #2
The manuscript entitled “Characterization and Antibacterial Activity of Silver Nanoparticles Synthesized from Oxya chinensis sinuosa (Grasshopper) Extract » is devoted to silver nanoparticles synthesize using a green method its characterization and antibacterial activity assessment against common foodborne pathogens, including Salmonella Typhimurium, Escherichia coli, Staphylococcus aureus, and Bacillus cereus. The experiments were conducted at a high methodological level, which does not allow us to doubt the results obtained. To my mind this manuscript is very topical and contains a lot of important and new data. It corresponding to the aims and scopes of the “Microorganisms” journal. I have no complaints about the scientific content of the material. I am ready to recommend it for publication after correcting several comments.
Comment 2-1:
It is worth indicating the conditions of TEM and SEM shooting
Response 2-1:
We sincerely thank you for your thoughtful suggestions. As per your advice, we have added the TEM and SEM measurement conditions to the Materials and Methods section, as shown below.
Revised sentence in line 144-146, page 3 in the revised manuscript.
The morphology and size of the AgNPs were examined using a biological transmission electron microscope (Bio-TEM) equipped with an H-7650 electron microscope (Hitachi, Japan) at an operating voltage of 120 kV.
Revised sentence in line 150-151, page 4 in the revised manuscript.
The morphology of the produced AgNPs was further analyzed using FE-SEM (SUPRA 40VP, Carl Zeiss, Germany) operated at 15 kV.
Comment 2-2:
It is worth indicating how the samples for FTIR were prepared
Response 2-2:
Following your suggestion, we have clearly specified the FTIR measurement conditions in the manuscript.
Added in line 170-172, page 4 in the revised manuscript.
For FTIR analysis, the washed O_AgNPs sample was freeze-dried to obtain a powder. The powdered O_AgNPs were then pressed into KBr pellets, and the spectra were recorded in the range of 400-4000 cm⁻¹.
Comment 2-3:
I would advise moving the description of the synthesis method from the results to the methods section
Response 2-3:
Thank you for your suggestion regarding the placement of the synthesis method. While we understand the intent behind your recommendation, we believe that retaining the synthesis method in the Results section offers essential context for interpreting the data and ensures a cohesive narrative throughout the manuscript. This structure allows for a more consistent interpretation. However, we are open to further discussion if adjustments are deemed necessary.
Comment 2-4:
The quality of Figure 2 is low, it looks like raw data, it should be moved to the supplementary
Response 2-4:
Thank you for your thorough feedback. As we fully agree with your suggestion, so Figure 2 was moved to the supplementary data.
Revised sentence in line 209-211, page 5 in the revised manuscript.
The high dispersion stability of the synthesized O_AgNPs was further confirmed via Tur-biscan analysis, which provided additional evidence of stability along with absorbance measurements (Figure S1).
Comment 2-5:
Redo the legend to Figure 3C, the captions should be clearer
Response 2-5:
In response to your suggestion, we have revised Figure 3C to ensure clearer distinction between the data.
Figure 3 in the original manuscript
Figure 3 in the revised manuscript
